# Grid-independent High Resolution Dust Emissions (v1.0) for Chemical Transport Models: Application to GEOS-Chem (12.5.0)

Jun Meng[1,2], Randall V. Martin[2,1,3], Paul Ginoux[4], Melanie Hammer[2,1], Melissa P. Sulprizio[5],

David A. Ridley[6], Aaron van Donkelaar[1,2]

[1]Department of Physics and Atmospheric Science, Dalhousie University, Halifax, Nova Scotia, B3H 4R2, Canada

[2] Department of Energy, Environmental & Chemical Engineering, Washington University in St. Louis, St. Louis, Missouri 63130, United States

[3]Smithsonian Astrophysical Observatory, Harvard-Smithsonian Center for Astrophysics, Cambridge, MA 02138, USA

[4]NOAA Geophysical Fluid Dynamics Laboratory, Princeton, New Jersey 08540, USA

[5]School of Engineering and Applied Science, Harvard University, Cambridge, MA 02138, USA

[6]California Environmental Protection Agency, Sacramento, CA 95814, USA

*Correspondence to*: Jun Meng (jun.meng@ucla.edu)

**Abstract.** The nonlinear dependence of the dust saltation process on wind speed poses a

challenge for models of varying resolutions. This challenge is of particular relevance for the next

generation of chemical transport models with nimble capability for multiple resolutions. We

develop and apply a method to harmonize dust emissions across simulations of different

resolutions by generating offline grid independent dust emissions driven by native high

resolution meteorological fields. We implement into the GEOS-Chem chemical transport model

a high resolution dust source function to generate updated offline dust emissions. These

updated offline dust emissions based on high resolution meteorological fields strengthen dust

emissions over relatively weak dust source regions, such as in southern South America,

southern Africa and the southwestern United States. Identification of an appropriate dust

emission strength is facilitated by the resolution independence of offline emissions. We find

that the performance of simulated aerosol optical depth (AOD) versus measurements from the

AERONET network and satellite remote sensing improves significantly when using the updated

offline dust emissions with the total global annual dust emission strength of 2,000 Tg yr$^{-1}$ rather

than the standard online emissions in GEOS-Chem. The updated simulation also better

represents in situ measurements from a global climatology. The offline high resolution dust

emissions are easily implemented in chemical transport models. The source code and global

offline high-resolution dust emission inventory are publicly available.

## 1 Introduction

Mineral dust, as one of the most important natural aerosols in the atmosphere, has significant impacts on weather and climate by absorbing and scattering solar radiation (Bergin et al., 2017; Kosmopoulos et al., 2017), on atmospheric chemistry by providing surfaces for heterogeneous reaction of trace gases (Chen et al., 2011; Tang et al., 2017), on the biosphere by fertilizing the tropical forest (Bristow et al., 2010; Yu et al., 2015) and ocean (Jickells et al., 2005; Guieu et al., 2019; Tagliabue et al., 2017), and on human health by increasing surface fine particulate matter ($PM_{2.5}$) concentrations (De Longueville et al., 2010; Fairlie et al., 2007; Zhang et al., 2013). Dust emissions are primarily controlled by surface wind speed to the third or fourth power, vegetation cover and soil water content. The principal mechanism for natural dust emissions is saltation bombardment (Gillette and Passi, 1988; Shao et al., 1993), in which sand-sized particles creep forward and initiate the suspension of smaller dust particles when the surface wind exceeds a threshold. The nonlinear dependence of dust emissions on meteorology introduces an artificial dependence of simulations upon model resolution (Ridley et al., 2013). For example, dust emissions in most numerical models are parameterized with an empirical method (e.g. Ginoux et al., 2001; Zender et al., 2003), which requires a critical wind threshold to emit dust particles. Smoothing meteorological fields to coarse resolution can lead to wind speeds falling below the emission threshold in regions that do emit dust. Methods are needed to address the artificial dependence of simulations upon model resolution that arises from nonlinearity in dust emissions.

Addressing this nonlinearity is especially important for the next generation of chemistry transport models that is emerging with nimble capability for a variety of resolutions at the

global scale. For example, the high performance version of GEOS-Chem (GCHP) (Eastham et al., 2018) currently offers simulation resolutions that vary by over a factor of 100 from C24 (~ 4°x4°) to C360 (~0.25°x0.25°), with progress toward even finer resolution and toward a variable

stretched grid capability (Bindle et al., 2020). Resolution-dependent mineral dust emissions would vary by a factor of 3 from C360 to C24 (Ridley et al., 2013). Such large resolution-dependent biases would undermine applications of CTMs to assess dust effects, and would lead to large within-simulation inconsistency for stretched grid simulations that can span the entire resolution range simultaneously. Grid-independent high resolution dust emissions offer a

potential solution to this issue.

An important capability in global dust evaluation is ground-based and satellite remote sensing. The Aerosol Robotic Network (AERONET), a global ground-based remote sensing aerosol monitoring network of Sun photometers (Holben et al., 1998), has been widely used to evaluate dust simulations. Satellite remote sensing provides additional crucial information

across arid regions where in-situ observations are sparse (Hsu et al., 2013). Satellite aerosol retrievals have been used extensively in previous studies to either evaluate the dust simulation (Ridley et al., 2012, 2016) or constrain the dust emission budget (Zender et al., 2004). Satellite aerosol products have been used to identify dust sources worldwide (Ginoux et al., 2012; Schepanski et al., 2012; Yu et al., 2018), especially for small-scale sources (Gillette, 1999).

The objective of this study is to develop a method to mitigate the large inconsistency of total dust emissions across different resolutions of simulations by generating and archiving offline dust emissions using native high resolution meteorological fields. We apply this method to the GEOS-Chem chemical transport model. As part of this effort, we implement an updated

high resolution satellite-identified dust source function into the dust mobilization module of
GEOS-Chem to better represent the spatial structure of dust sources. We apply this new
capability to assess the source strength that best represents observations.

## 2 Materials and Methods

### 2.1 Description of Observations

We use both ground-based and satellite observations to evaluate our GEOS-Chem simulations.
AERONET is a global ground-based remote sensing aerosol monitoring network of sun
photometers with direct sun measurements every 15 minutes (Holben et al., 1998). We use
Level 2.0 Version 3 data that has improved cloud screening algorithms (Giles et al., 2019).
Aerosol optical depth (AOD) at 550 nm is interpolated based on the local Angstrom exponent at
the 440 nm and 670 nm channels.

Twin Moderate-Resolution Imaging Spectroradiometer (MODIS) instruments aboard
both on the Terra and Aqua NASA satellite platforms provide near daily measurements globally.
We use the AOD at 550 nm retrieved from Collection 6.1 (C6) of MODIS product (Sayer et al.,
2014). We use AOD from the Deep Blue (DB) retrieval algorithm (Hsu et al., 2013; Sayer et al.,
2014) designed for bright surfaces, and the Multi-Angle Implementation of Atmospheric
Correction (MAIAC) algorithm (Lyapustin et al., 2018), which provides global AOD retrieved
from MODIS C6 radiances at a resolution of 1 km. The MAIAC AOD used in this study is
interpolated to the AOD value at 550 nm.

We use ground-based surface fine dust concentration measurements over the US from
the Interagency Monitoring of Protected Visual Environments (IMPROVE,

http://vista.cira.colostate.edu/Improve/ ) network. The IMPROVE network provides 24-hr average fine dust concentrations data every third day over the national parks in the United States. We also include a climatology of dust surface concentrations measurements over 1981-2000 from independent dust measurement sites over the globe (Kok et al. 2020). We use those

sites (12 in total) (Figure S1) that are either in the dust belt across Northern Hemisphere or sites relatively close to the weak emission regions in the Southern Hemisphere to evaluate our dust simulation.

We compare the simulated AOD and dust concentrations with measurements using reduced major axis linear regression. We report root mean square error (E), correlation (R) and

slope (M).

## 2.2 Dust mobilization module

We use the dust entrainment and deposition (DEAD) scheme (Zender et al., 2003) in the GEOS-Chem model to calculate dust emissions. The saltation process is dependent on the critical

threshold wind speed, which is determined by surface roughness, soil type and soil moisture. Dust aerosol is transported in four size bins (0.1-1.0, 1.0-1.8, 1.8-3.0, and 3.0-6.0 $\mu$m radius). Detailed description of the dust emission parameterization is in Sect. S1 of the supplemental material.

The fractional area of land with erodible dust is represented by a source function. The

dust source function used in the dust emission module plays an important role in determining the spatial distribution of dust emissions. The standard GEOS-Chem model (version 12.5.0) uses a source function at 2° x 2.5° resolution from Ginoux et al. (2001) as implemented by Fairlie et

al. (2007). We implement an updated high resolution version of the dust source function in this study at 0.25° x 0.25° resolution (Sect. S2). Figure S2 shows a map of the original and updated

version of the dust source function. The updated source function exhibits more spatially resolved information due to its finer spatial resolution resulting in a higher fraction of erodible dust over in the eastern Arabian Peninsula, the Bodélé depression, and the central Asian deserts. The dust module dynamically applies this source function, together with information on soil moisture, vegetation, and land use to calculate hourly emissions using the Harmonized

Emissions Component (HEMCO) module described below.

**2.3 Offline dust emissions at the native meteorological resolution**

HEMCO (Keller et al., 2014) is a stand-alone software module for computing emissions in global atmospheric models. We run the HEMCO standalone version using native meteorological

resolution (0.25° x 0.3125°) data for wind speed, soil moisture, vegetation, and land use to archive the offline dust emissions at the same resolution as the meteorological data. The computational time required for calculating offline dust emission fluxes at  0.25° x 0.3125° resolution is around 6 hours for one-year of offline dust emissions on a compute node with 32 cores on 2 Intel CPUs at 2.1 GHz. In this study, we generate two offline dust emission datasets

at 0.25° x 0.3125° resolution. One, referred to as the default offline dust emissions, uses the existing dust source function in the GEOS-Chem dust module; the other, referred to as the updated offline dust emissions, uses the updated dust source function implemented here. Both datasets are at the hourly resolution of the parent meteorological fields. The archived native resolution offline dust emissions can be conservatively regridded to coarser resolution for

consistent input to chemical transport models at multiple resolutions. We use the GEOS-Chem

model to evaluate the dust simulations and the emission strength.

**2.4 GEOS-Chem chemical transport model and simulation configurations**

GEOS-Chem (Bey et al., 2001; The International GEOS-Chem User Community, 2019) is a three-

dimensional chemical transport model driven by assimilated meteorological data from the

Goddard Earth Observation System (GEOS) of the NASA Global Modelling and Assimilation

Office (GMAO). The GEOS-Chem aerosol simulation includes the sulfate-nitrate-ammonium

(SNA) aerosol system (Fountoukis and Nenes, 2007; Park et al., 2004), carbonaceous aerosol

(Hammer et al., 2016; Park et al., 2003; Wang et al., 2014), secondary organic aerosols (Marais

et al., 2016; Pye et al., 2010), sea salt (Jaeglé et al., 2011) and mineral dust (Fairlie et al., 2007)

with updates to aerosol size distribution (Ridley et al., 2012; Zhang et al., 2013). Aerosol optical

properties are based on the Global Aerosol Data Set (GADS) as implemented by Martin et al.

(2003) for externally mixed aerosols as a function of local relative humidity with updates based

on measurements (Drury et al., 2010; Latimer and Martin, 2019). Wet deposition of dust,

including the processes of scavenging from convection and large scale precipitation, follows Liu

et al. (2001).  Dry deposition of dust includes the effects of  gravitational settling and turbulent

resistance to the surface, which are represented with deposition velocities in the

parameterization, implemented into GEOS-Chem by Fairlie et al. (2007).

        The original GEOS-Chem simulation used online dust emissions by coupling the dust

mobilization module online. We develop the capability to use offline dust emissions based on

the archived fields described in Sect. 2.3. We conduct global simulations with GEOS-Chem

(version 12.5.0) at a horizontal resolution of 2° by 2.5° for the year 2016. Simulations using the online and offline dust emissions are conducted to evaluate the offline dust emissions. We conduct two simulations using online dust emissions with different dust source functions. The

first is with the original version of the dust source function, hereafter noted as the original online dust simulation. The other is with the updated version of source function, in which the updated fine resolution source function is interpolated to 2° by 2.5° resolution. The annual total emissions for the online dust emissions are at the original value of 909 Tg yr$^{-1}$. We conduct another four sets of simulations using offline dust emissions. The first uses the default offline

dust emissions with annual total dust emission of 909 Tg yr$^{-1}$. The remaining use the updated offline dust emissions with the annual total dust emission scaled to 1,500, 2,000 and 2,500 Tg yr$^{-1}$, which are in the range of the current dust emission estimates of over 514 – 4,313 Tg yr$^{-1}$ (Huneeus et al., 2011). We focus on the simulation with 2,000 Tg yr$^{-1}$ which better represents observations as will be shown below.


## 3 Results and Discussion

### 3.1 Spatial and seasonal variation of the offline dust emissions

Figure 1 shows the spatial distribution of the annual and seasonal dust emission flux rate for the updated offline dust emissions. The annual dust emission flux rate is high over major

deserts, such as the northwestern Sahara, the Bodélé Depression in northern Chad, eastern Arabian Peninsula and central Asian Taklimakan and Gobi deserts. There are also hotspots of dust emission flux rate over relatively smaller deserts, such as the Mojave Desert of the southwestern United States, the Atacama desert of southern South America, the Kalahari

desert on the west coast of southern Africa and the deserts of central Australia. Those features

reflect the fine resolution of the updated dust source function and of the offline dust emissions.

Seasonally, the dust emission flux rate resembles the annual distribution, however, with a

lower dust emission flux rate over the Bodélé Depression in northern Chad in summer and

higher dust emission flux rate over the Middle East and central Asian deserts in spring and

summer.

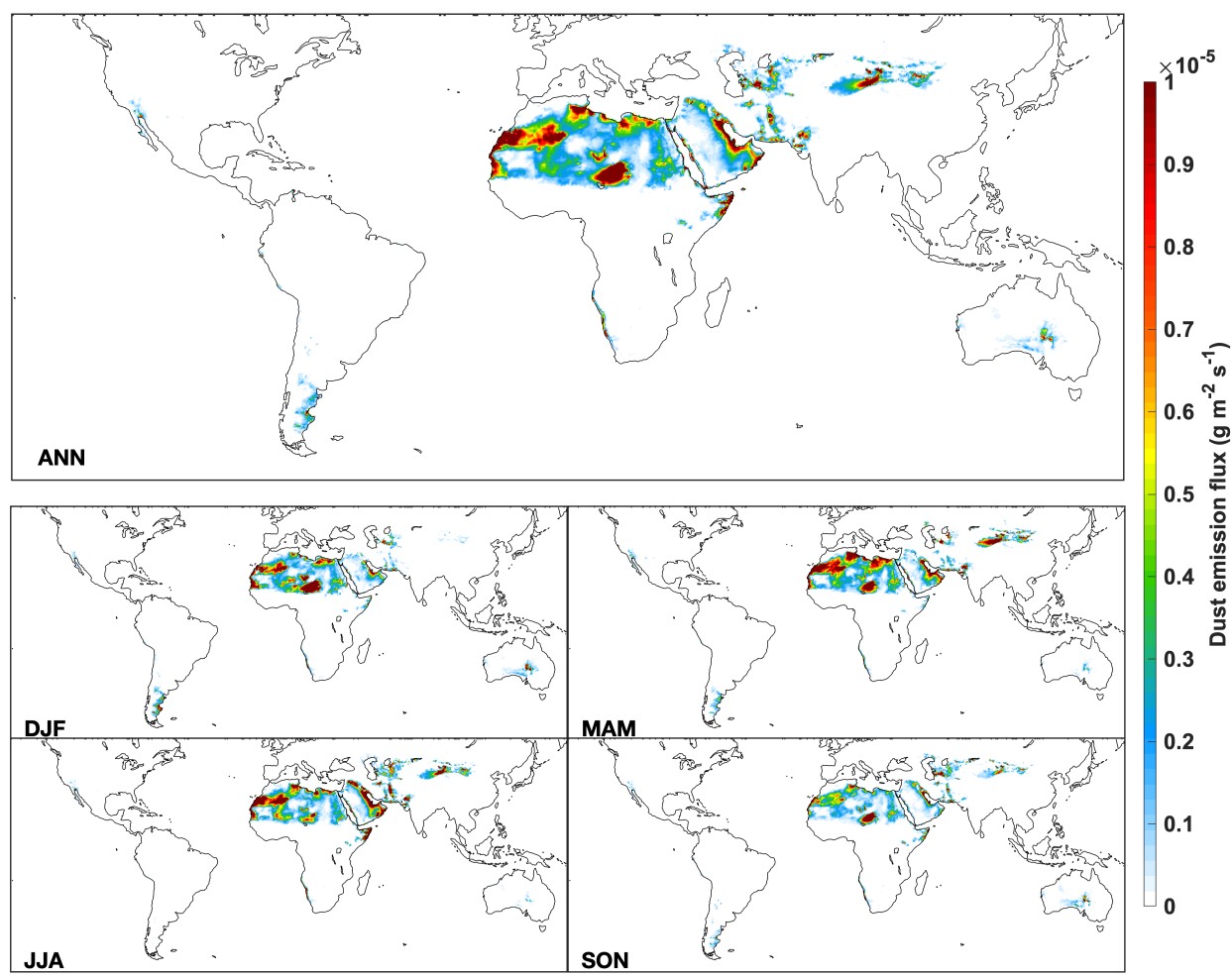


Figure 1. Annual and seasonal mean dust emission flux rate for the offline high resolution dust emissions with updated dust source function and updated annual total dust emission of 2,000 Tg.

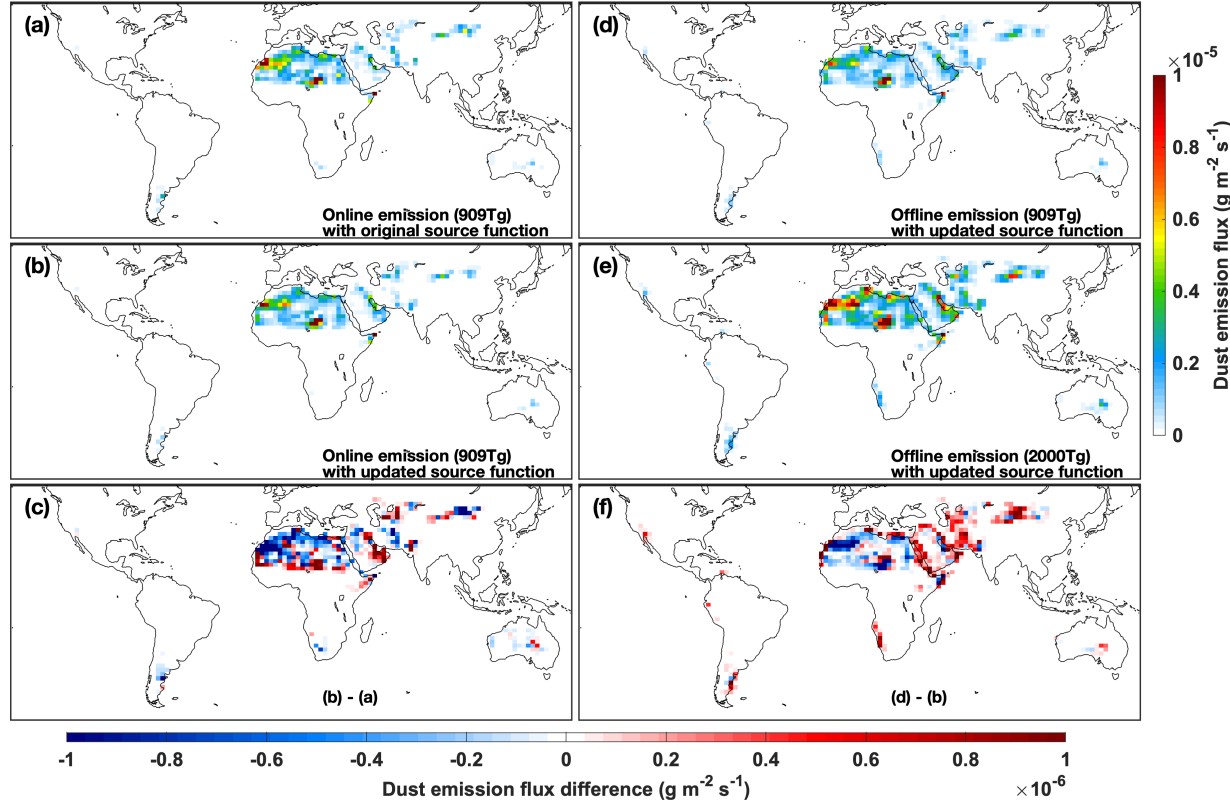

Figure 2. Annual mean dust emission flux rate for 2016. (a) The original online dust emissions with original dust source function and annual total dust emissions of 909 Tg. (b) Online dust emissions with updated dust source function. (c) Difference of flux rate between online dust emissions using original and updated dust source functions. (d) Offline dust emissions with updated dust source function. (e) Offline dust emissions with updated dust source function and updated annual total dust emissions of 2,000 Tg. (f) Difference of flux rate between offline and online dust emissions. The online dust emissions are in 2° x 2.5° resolution. The offline dust emissions shown in (b), (d), (f) are regridded from 0.25° x 0.3125° resolution to 2° x 2.5° for comparison with online dust emissions.

Figure 2 shows the spatial distribution of the annual dust emission flux rate for the

online and offline dust emissions with the original and updated dust source functions with

original and updated global total dust source strengths. All simulations exhibit high dust

emission flux rates over major desert regions, such as the North African, Middle East and

Central Asian deserts, with local enhancements over the western Sahara and northern Chad.

The simulation with the updated source function exhibits stronger emissions in the Sahara and

Persian Gulf regions (Fig. 2c). The difference between the online and offline dust emissions,

shown in Fig. 2f, can be considered the error in the online approach arising from coarse

resolution meteorological fields. The offline dust emissions based on native resolution

meteorological fields have lower dust emission flux rates over northwest Africa, but higher dust

emission flux rates over the Middle East and Central Asia. Higher annual dust emission flux

rates over the southwestern United States, southern South America, the west coast of southern

Africa and central Australia in the offline dust emissions reflect that the native resolution offline

dust emissions are strengthened over relatively weaker dust emission regions. Generally,

coastal and minor desert regions emit more dust when calculating emissions at the native

meteorological resolution.

Figures S3–S6 show the seasonal variations of dust emission flux rates for online and

offline emissions. The offline dust emissions have lower emission flux rates than the online dust

emissions during spring (March, April and May) (MAM) and winter (December, January and

February) (DJF) over North Africa. The offline dust emission flux rate is higher than the online

dust emission flux rate over the Middle East and Central Asian deserts during spring and

summer (June, July and August) (JJA). Emission flux rates are low over Central Asian deserts

during winter. The strengthening of offline dust emissions over weaker dust emitting regions

persists throughout all seasons.

**3.2 The performance of AOD simulations over desert regions**

Figure 3 shows simulated AOD using the updated offline dust emissions. Difference maps of

simulated AOD between online and offline dust emissions are shown in Figure S7. We select for

evaluation the AERONET sites where the ratio of simulated dust optical depth (DOD) to

simulated total AOD exceeds 0.5 in the simulation using the updated offline dust emissions with

annual dust strength of 2,000 Tg. Annually, the simulated DOD has the highest value over the

Bodélé Depression. This feature persists in all seasons except summer when DOD has the

245 highest values over the western Sahara and eastern Arabian Peninsula. The scatter plots show

that annually the simulated AOD from both simulations are highly correlated with AERONET

measurements across the dust regions (R = 0.86-0.88). The simulation with updated offline dust

emissions has an improved slope and smaller root mean square error than the simulation using

the original online dust emissions. AOD from the simulation with updated offline dust emissions

250 is also more consistent with the measurements in different seasons, especially in the spring

(MAM) and fall (SON) with slopes close to unity and R exceeding 0.9.

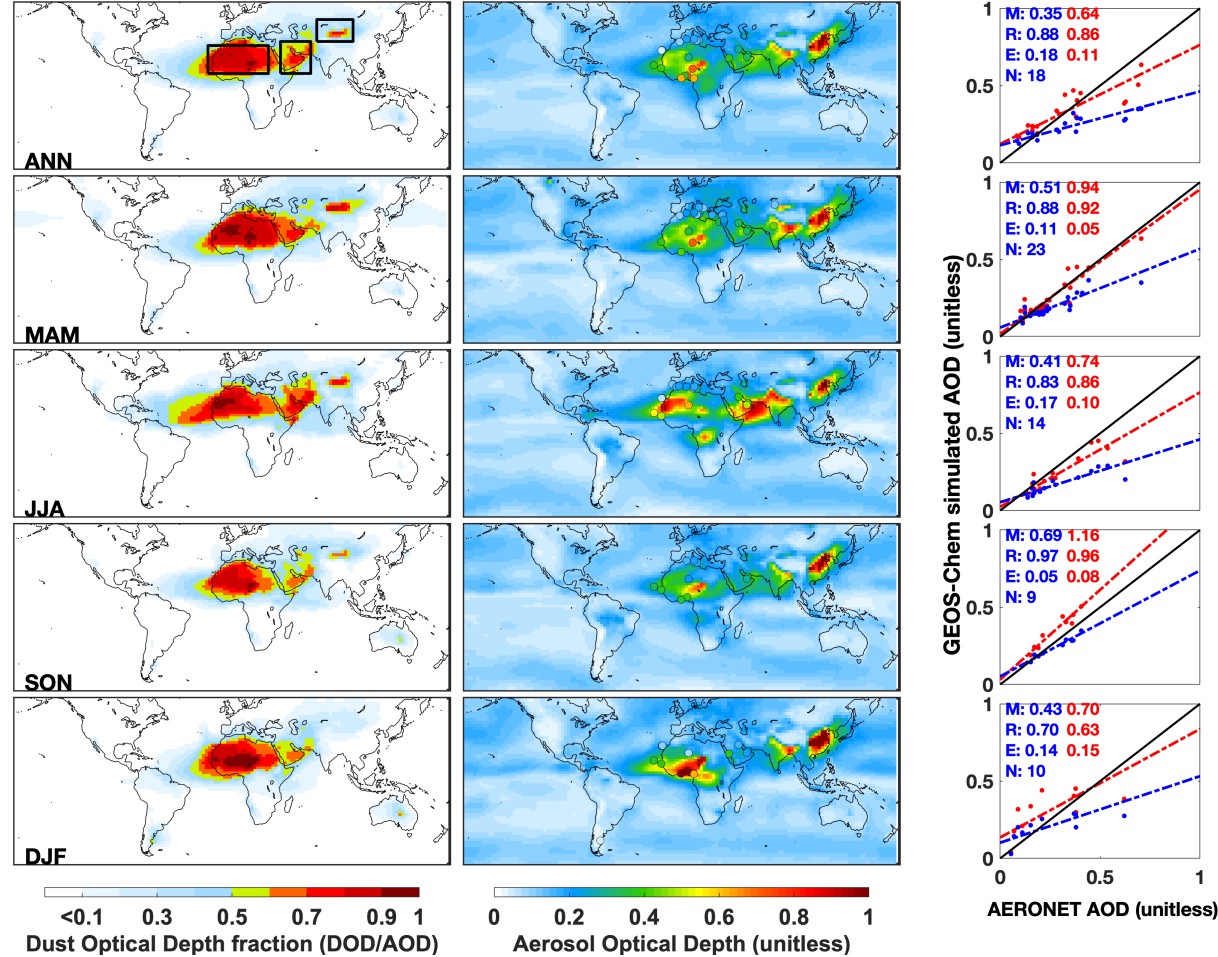

Figure 3. Annual and seasonal mean simulated dust optical depth (DOD) fraction (left column) and aerosol optical depth (AOD) (middle column) from GEOS-Chem simulations for 2016, and AERONET measured AOD at sites where the ratio of simulated DOD and AOD exceeds 0.5, which are shown as filled circles in the middle column. Boxes in the left top panel outline the three major deserts examined in Figure 4. The right column shows the corresponding scatter plot with root mean square error (E), correlation coefficient(R) and slope (M) calculated with reduced major axis linear regression. N is the number of valid ground-based monitoring records. The results for the simulation using the original dust emissions are shown in blue; the results for the simulation using updated dust emissions with dust strength of 2,000 Tg yr$^{-1}$ are shown in red. The best fit lines are dashed. The 1:1 line is solid.

We further evaluate the performance of simulated AOD over major desert regions using the MODIS Deep Blue (DB) and MAIAC AOD products. Figure 4 shows annual and seasonal scatter plots comparing GEOS-Chem simulated AOD using original online dust emissions and updated offline dust emissions against retrieved AOD from MODIS DB and MAIAC satellite

products over the three major desert regions outlined in Fig. 3. Figure S8 shows the annual and

seasonal AOD distribution from MODIS DB and MAIAC. Annually, the simulation using updated

offline dust emissions exhibits greater consistency with satellite AOD than does the simulation

using original online dust emissions across all three desert regions. The simulation using

updated offline dust emission performs better across all three desert regions and in all four

seasons except for North Africa in summer, during which AOD is overestimated. Both

simulations underestimate AOD over central Asian deserts during winter when dust emissions

are low and other sources may be more important. Overall, the simulation using original online

dust emissions underestimates AOD over all three major desert regions, especially over the

Middle East and central Asian deserts. The simulation using updated offline dust emissions

exhibits greater consistency with satellite observations with higher slopes and correlations.

**3.3 Evaluation of the simulations against surface dust concentration measurements**

We also evaluate our simulations using different dust emissions against measurements of

surface dust concentrations. Figure 5 shows the comparison of modeled fine dust surface

concentration against the fine dust concentration observation from the IMPROVE network. The

simulations using the updated offline dust emissions can better represent the observed surface

fine dust concentration measurements than the simulation using the original online dust

emissions with higher correlations and slopes across all seasons. Annually, the correlation

between the simulation and observation increases from 0.39 to 0.68 , and the slope increases

from 0.31 to 0.71 when using the updated offline dust emissions with annual dust strength of

909 Tg compared to the simulation using the original online dust emissions. Scaling the annual

dust strength to 2,000 Tg/yr marginally improves the performance of the model simulation of

fine dust concentrations in all seasons except winter, during which the surface fine dust

concentrations are overestimated. Given the specificity and density of the dust measurements,

and the disconnect of North American dust emissions from the global source, we conduct an

additional sensitivity simulation with North American dust emissions reduced by 30%. The right

column shows that the annual slope in the resultant simulation versus observations improves

to 1.07, minor improvements to annual and seasonal correlations. Future efforts should focus

on better representing the seasonal variation of dust emissions.

Figure 6 shows the comparison of seasonal averaged modeled and measured surface

dust concentrations from 12 independent sites across the globe. The simulation using the

updated offline dust emissions with dust strength of 2,000 Tg yr$^1$ is more consistent with the

observations at almost all sites. The remaining bias at sites distant from source regions, for

example sites in the Southern Hemisphere and East Asia, likely reflects remaining uncertainty in

representing dust deposition. Further research is needed to address remaining knowledge gaps,

such as better representing the dust size distribution and deposition during transport.


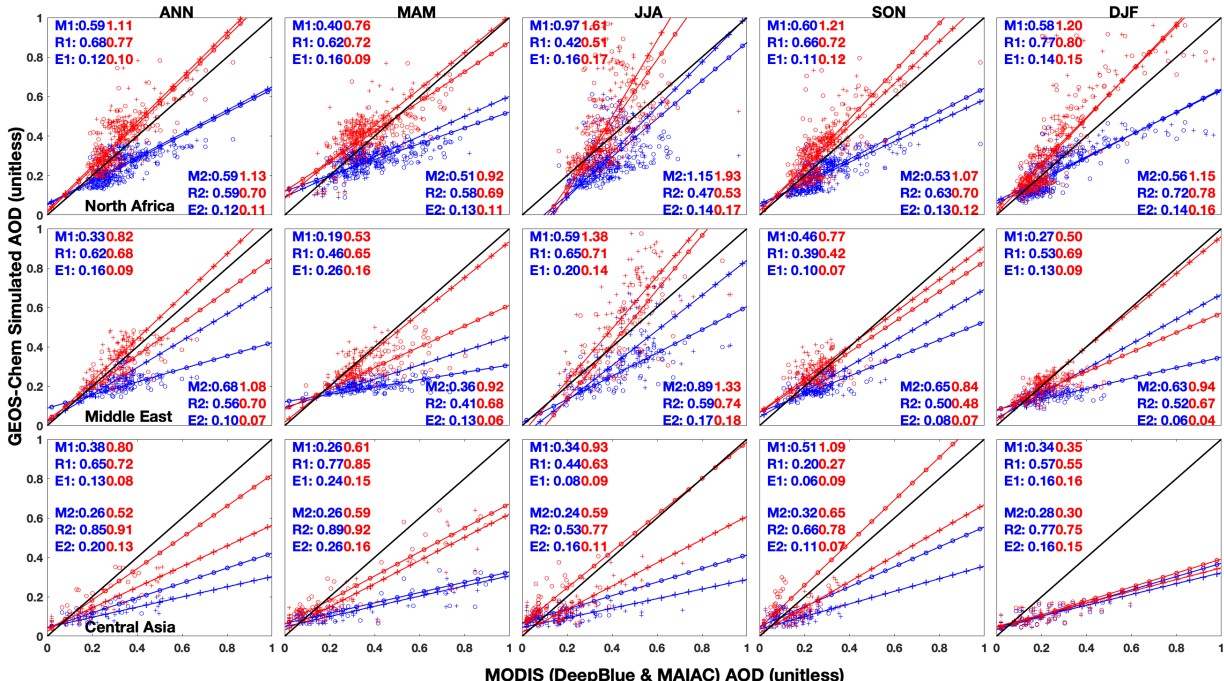

Figure 4. Scatter plots and statistics of comparing GEOS-Chem simulated AOD with satellite AOD over desert regions annually (the first column) and seasonally (the right four columns). The results for the North African, Middle East and Central Asia deserts are shown in the top, middle and bottom rows
respectively. The results for the simulation using the original dust emissions are shown in blue; the results for the simulation using updated dust emissions with dust strength of 2,000 Tg yr$^{-1}$ are shown in red. Open circles represent the comparison with MODIS Deep Blue AOD; the plus signs represent the comparison with MAIAC AOD. Correlation coefficient(R), root mean square error (E), and Slope (M) are reported, in which R1, E1 and M1 show the results of the comparison with MODIS Deep Blue AOD; R2,
E2 and M2 show the results of the comparison with MAIAC AOD. The best fit lines are dashed lines with corresponding marker signs and colors. The 1:1 line solid black line.

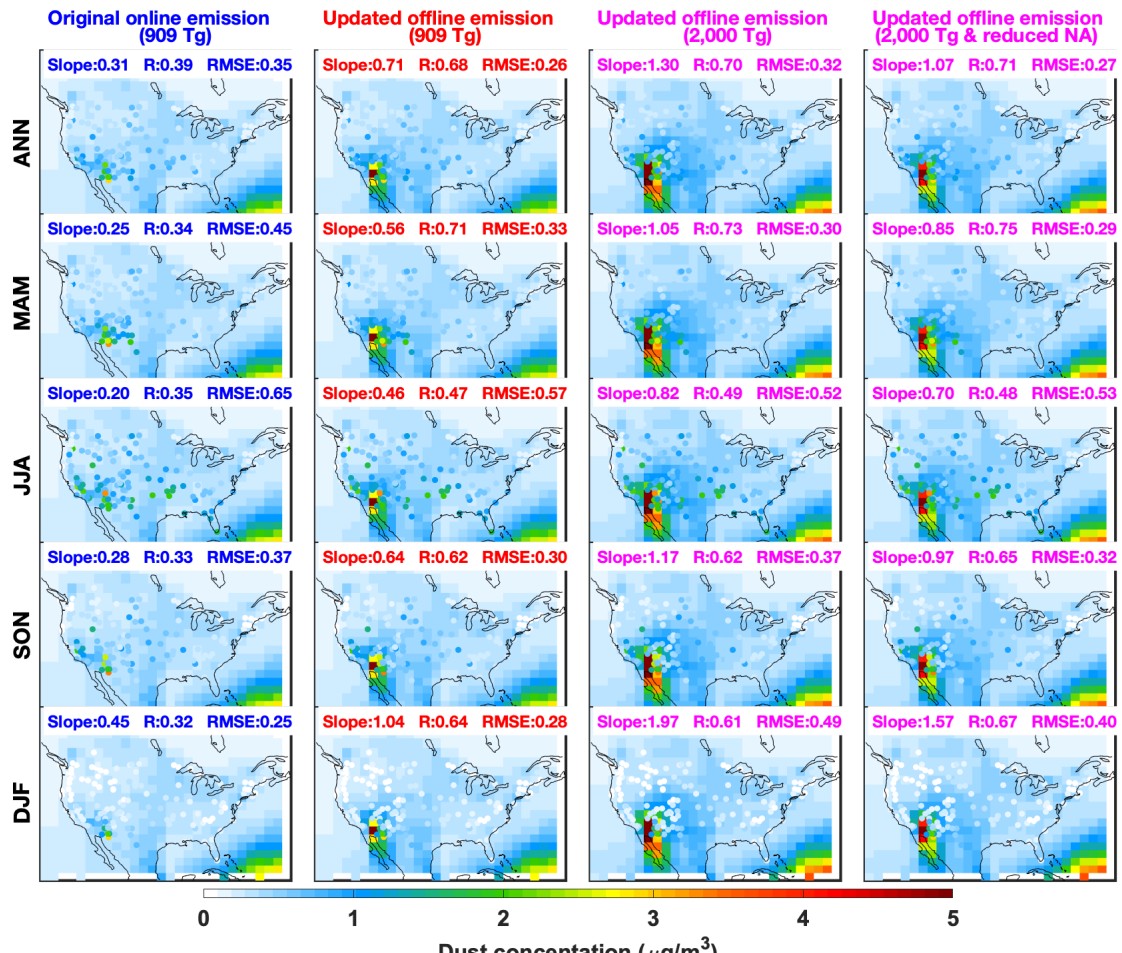

Figure 5. Annual and seasonal mean simulated fine dust concentrations from GEOS-Chem simulations with different dust emissions for 2016, and IMPROVE fine dust measurements, which are shown as filled circles. Root mean square error (E), correlation coefficient(R) and slope (M) calculated with reduced major axis linear regression are reported. The results for the simulation using the original dust emissions are shown in blue (left column); the results for the simulation using updated dust emissions with dust strength of 909 Tg yr$^{-1}$ are shown in red (second column); the results for the simulation using updated dust emissions with dust strength of 2,000 Tg yr$^{-1}$ are shown in magenta (third column); the right column is the sensitivity simulation with North America dust emission reduced by 30%.

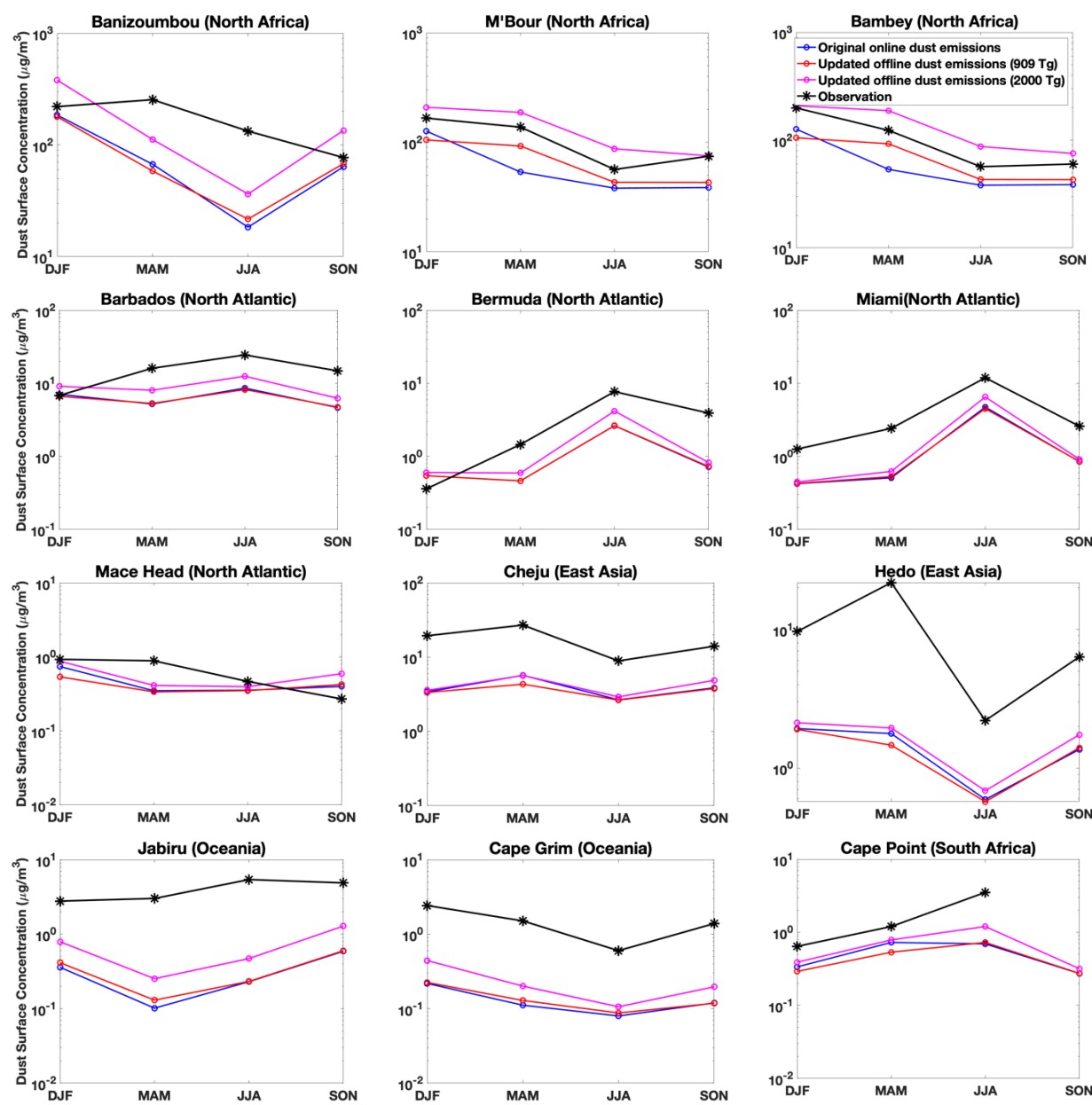

Figure 6. Comparison of modeled and measured seasonal averaged surface dust concentrations at 12 independent globally distributed sites for the years 1981-2000. Nine sites are in the dust belt across Northern hemisphere. The remaining 3 sites are relatively close to the weak dust emission regions in Southern Hemisphere. The results for the simulation using the original dust emissions are shown in blue; the results for the simulation using updated dust emissions with dust strength of 909 Tg yr[-1] are shown in red; the results for the simulation using updated dust emissions with dust strength of 2,000 Tg yr[-1] are shown in magenta. The measurements are in black.

**3.4 Discussion of the dust source strength**

One of the advantages of the offline dust emissions is that the same dust source strength can

be readily applied to all model resolutions, facilitating evaluation of dust source strength

independent of resolution. We have found that the simulation with global total annual dust

emission scaled to 2,000 Tg better represents observations than does the default simulation

with global total annual dust emissions of 909 Tg. We also evaluate simulations with global total

annual dust emission scaled to 1,500 Tg and 2,500 Tg. Figure S9 indicates that the simulation

with global total annual dust emission scaled to 2,000 Tg is more consistent with satellite

observations over North Africa and the Middle East. Although the central Asian deserts and

regions with AERONET observations (Fig. S10) are better represented by the simulation with

global total annual dust emission scaled to 2,500 Tg, since North Africa has the highest dust

emissions (Huneeus et al., 2011), and AOD over North Africa is most likely dominated by dust,

we scale global total annual dust emissions to best match this source region robustly. We

refrain from applying a regional scale factor to the central Asian deserts given the paucity of in

situ measurements. More dust-specific observations are needed to constrain dust emissions for

the Asian deserts region and other deserts. Additional development and evaluation should be

conducted to further narrow the uncertainty of dust emissions, especially at the regional scale.

Although the main purpose of this manuscript is to develop and evaluate an offline grid-

independent inventory, it is worth noting that online models have the capability to scale to a

target source strength. In that context the global source strength identified here may be of use

for global online models to scale to the global source strength, with the caveat that differences

in dust parameterization, dust optics, and deposition may affect performance.

**3.5 Advantages of high resolution offline dust emissions for model development**

Uncertainty remains in the estimated global annual total dust emissions. Direct dust emission flux observations are few. Current atmospheric models apply a global scale factor to optimize with a specific set of ground observations. Because of the non-linear dependence on resolution of the dust emissions, the source strength has historically depended upon model resolution, which inhibits general evaluation. The native resolution offline dust emissions facilitate consistent evaluation and application across all model resolutions. Such consistency is particularly important for stretched-grid simulations with the capability for factors of over 100 variation in resolution within a single simulation (Bindle et al., 2020).

**4 Summary and Conclusions**

The nonlinear dependence of dust emission parameterizations upon model resolution poses a challenge for the next generation of chemical transport models with nimble capability for multiple resolutions. The method explored here to calculate offline dust emissions at native meteorological resolution promotes consistency of dust emissions across different model resolutions. We take advantage of the capability of HEMCO standalone module to calculate dust emission offline at native meteorological resolution using the DEAD dust emission scheme combined with an updated high resolution dust source function. We evaluate the performance of the simulation with native resolution offline dust emissions and an updated dust source function with source strength of 2,000 Tg yr$^{-1}$. We find better agreement with measurements, including satellite and AERONET AOD, and surface dust concentrations. The offline fine resolution dust emissions strengthen the dust emissions over smaller desert regions. The

independence of source strength from simulation resolution facilitates evaluation with

observations. Sensitivity simulations with an annual global source strength of either 1,500 or

2,500 Tg generally degraded the performance. A sensitivity simulation with North American

emissions reduced by 30% improved the annual mean slope versus observations. Future work

should continue to develop and evaluate the representation of dust deposition and regional

seasonal variation.

## 5 Code and Data Availability

The source code for generating the offline dust emissions is available on GitHub

(https://github.com/Jun-Meng/geos-chem/tree/v11-01-Patches-UniCF-vegetation) and Zenodo

repository ( https://doi.org/10.5281/zenodo.4062003 ) (Meng et al., 2020b). The instruction of

how to generate the emission files is in the README.md file in the GitHub repository

(https://github.com/Jun-Meng/geos-chem/tree/v11-01-Patches-UniCF-vegetation). The global

high resolution (0.25°x0.3125°) dust emission inventory is available on Zenodo

(https://doi.org/10.5281/zenodo.4060248) (Meng et al., 2020a), containing netCDF format files

of global gridded hourly mineral dust emission flux rate. Currently, the dataset (version1.0) is

available for the year 2016. The dataset for other years since 2014 will be available in future

versions.

The base GEOS-Chem source code in version 12.5.0 is available on Github

(https://github.com/geoschem/geos-chem/tree/12.5.0) and Zenodo repository

(https://zenodo.org/record/3403111#.X7PKv5NKiF0,%202019). The GEOS-Chem simulation

output data and AOD observations used to evaluate the model performance, including MODIS

Deep Blue, MODIS MAIAC and AERONET AOD, can be accessed via this Zenodo repository

([https://doi.org/10.5281/zenodo.4312944](https://doi.org/10.5281/zenodo.4312944)) (Meng et al., 2020c).

**Information about the Supplement**

The supplement related to this article describes the details of the dust emission scheme used in

this project, the updated high resolution dust source function, as well as additional figures

described in the main text.

**Author contributions**

RVM and JM conceived the project. JM developed the dust emission dataset using data and

algorithms from DAR, PG, MH, AvD, and MPS. JM prepared the manuscript with contributions

from all coauthors. All authors helped revise the manuscript.

**Competing interests**

The authors declare that they have no conflict of interest.

**Acknowledgement**

This work was supported by the Natural Science and Engineering Research Council of Canada.
Jun Meng was partially supported by a Nova Scotia Research and Innovation Graduate
Scholarship. Martin acknowledges partial support from NASA AIST-18-0011. We are grateful to
Compute Canada and Research Infrastructure Services in Washington University in St. Louis for
computing resources. The meteorological data (GEOS-FP) used in this study have been provided
by the Global Modeling and Assimilation Office (GMAO) at NASA Goddard Space Flight Center.
We thank Jasper Kok and Longlei Li for providing the compilation of independent surface dust
concentrations measurements. We thank the four anonymous reviewers for their constructive
comments and suggestions. All figures are produced with the MATLAB software.

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
