# Peer review of "Grid-independent High Resolution Dust Emissions (v1.0) for Chemical Transport Models: Application to GEOS-Chem (12.5.0)"

_Geoscientific Model Development, 2020_

## Referee Comment (RC1) · Anonymous Referee #1 · 8 Jan 2021

General comments:

In this paper, Meng et al. proposed a grid-independent dust emissions module for CTMs (GEOS-Chem in this case), where dust flux is being calculated a priori and offline within the emissions module (HEMCO in this case). In essence, this approach seems to be replacing the error due to interpolating meteorological fields with those due to interpolating the final dust flux. Considering nonlinear relations between meteorological fields and calculated dust fluxes, I am concerned about using this interpolated dust flux together with the interpolated meteorological fields later in a CTM to represent dust transport, diffusion, deposition, etc. The benefit of an online approach (other

than the aerosol feedback) is a physical consistency between wind, relative humidity, temperature, soil moisture, etc. representing both dust emissions and other phenomena such as dust transport and deposition, and I am not convinced that the offline approach provides the same faithfulness. Along that line, I found a number of major issues both with the proposed approach as well as its evaluation as summarized in the next section.

Specific comments:

1/ If the only major benefit of the offline approach is a better resolution, simulations should be conducted with various grid resolutions rather than just contrasting the two extremes (2deg x 2.5deg vs. 0.25deg x 0.3125deg). Specifically, comparisons should be conducted to ensure that the online and offline techniques indeed give the exact same dust flux with the same grid resolution.

2/ The soil moisture (another important meteorological factor) was not mentioned when using the offline approach. Wind and soil moisture are dynamically linked and should be represented and discussed.

3/ Based on the scatterplots in figures 3 and 4, the offline model seems to almost always give higher dust emissions, but figure 2(f) shows considerably lower values from the offline model over the Sahara and Sahel. Please discuss this inconsistency. Also, along that line, figure 3 should include a comparison between spatial distributions of online vs offline (overlaid with the AERONET obs) AOD. Perhaps replace the DOD/AOD column and move it to the Supplement?

4/ Throughout the abstract, main text, and conclusion sections, the offline model is argued to better resolve weak dust source regions, but no evaluations are provided for these regions.

5/ I see no connection (and in fact no scientific merit from the physics point of view) between the scaling factor and the offline approach. The scaling is not an advantage

of the offline model as described in sec. 3.3. An online model can also be scaled using the parameter "C" in Eq. (1) of the Supplement. Additionally, the paper misses the justification behind the chosen scaling factors as well as a detailed evaluation.

6/ Additional computational time required for calculating dust fluxes in HEMCO when using the offline approach should be presented and discussed.

---

## Referee Comment (RC2) · Anonymous Referee #2 · 14 Jan 2021

This work provides a quick and smart way to fix an issue associated with the nonlinear dependency of soil dust emissions on grid resolutions. Dust emissions are known to be a function of fourth or third power of grid-scale wind speeds, which are not a conserved quantity while regridding meteorological data for downscaling or upscaling simulations. As a result, this causes several issues, including not only simulated discrepancy from the observations but also more seriously a breach of mass conservation of soil dust aerosols using the same meteorology but different grid resolutions. Authors developed an offline approach to pre-calculate dust emissions with the finest wind speed data and to use it independent on the model grid resolutions. By doing this, they were able to use consistent dust emissions for simulations with different grid resolutions and also

even scale it to better match with observations. I recommend this work for publication at GMD after addressing a few concerns about this work as follows:

1. Description of soil dust mobilization using the finest wind speed data is clearly written but its application for the coarse model resolutions needs a bit of more elaboration. For example, dust emissions on a finer resolution are simply summed up online while used for coarse model simulations. If this is the case, an instantaneous mixing of dust emission would occur. I guess that this would not be a serious issue for fine-mode dust bins. But, for example, coarse-mode dust aerosols emitted from 0.2 degree grid box could be instantaneously mixed into 2 degree grid box. Any problem with this?

2. Following up the first comment above, how do you simulate the dry deposition of dust aerosols, which are also dependent on grid-scale mixing and other micro-scale meteorological parameters. A similar issue could arise if authors use the same grid-scale dry deposition calculation especially for coarse mode dust aerosols.

3. I wonder how the model evaluation against AOD observations could properly represent the model capability to simulate dust aerosols and especially dust mobilization. Not only AOD but also Angstrom exponent or other aerosol optical properties would be helpful to give a proper measure of the model performance.

---

## Referee Comment (RC3) · Anonymous Referee #3 · 25 Jan 2021

The authors generate high spatial and temporal resolution mineral dust emissions fields that can be prescribed for use in lower-resolution simulations with the GEOS-Chem model. Online dust emissions are well known to depend on model resolution because of nonlinearity in the governing parameterization. The use of consistent dust emissions across model resolutions overcomes this problem. I disagree with Referee 1 on his/her main point, and agree with Referee 3 on his/hers. The modeler wants to represent the most accurate dust emissions distribution possible irrespective of model resolution, and in particular whether or not they are consistent with the coarse model wind field. Representing high-resolution emissions, even crudely at lower resolution, is much preferred than not representing them at all. Specifically, smoothing of the wind fields at coarser

resolution leads to wind speeds falling below the threshold and zero dust emissions in locations that do emit dust. Scaling of global emissions to match those generated at higher resolution leads to unrealistic amplification (hotspots) elsewhere. This is a real problem that the proposed methodology alleviates.

The authors make good use of AERONET, MODIS-DB and MAIAC datasets to justify an annual global emission total for the year concerned.

The manuscript is well written and mostly clear. I recommend the manuscript be accepted with some clarifications.

1. Question: Please clarify how the high resolution (0.25 deg. x 0.25 deg.) satellite-identified dust source function (line 79, 110) is obtained. Section S1 refers to Ginoux et al (2001) and Zender et al. (2003). However, how are the surface factors S and in particular the Am factors obtained at this higher resolution? How is the updated source function then applied (presumably interpolated?) for online 2x2.5 deg. simulations? Please clarify.

2. Line 127: sounds like the default emissions would be at the 2 x 2.5 deg. resolution original source function. Please clarify.

3. Line 154: it is unclear if these 2 simulations are both conducted at 2x2.5 deg. resolution.

---

## Referee Comment (RC4) · Anonymous Referee #4 · 29 Jan 2021

The production of an improved dust emission dataset would be useful for CTMs varying spatial resolution (and idealized simulations in multi-model studies like AeroCom). Generation of high spatial and temporal resolution mineral dust emissions fields for use in lower-resolution simulations is thus a worthwhile product. However, I find the light evaluation of the simulations a concern. Improving the evaluation would increase confidence if the proposed dust emissions are representative at both global and regional levels. For example, only having a single AERONET observation site (and only in spring?) for all of Asia is particularly concerning, especially given issues with satellite retrievals over land. There also appears to be no evaluation of Southern Hemisphere sources. Until a more comprehensive evaluation of the dust atmospheric state is un-

dertaken an evaluation of emissions, and thus the dataset as whole, is difficult to judge.

It is probably good to point out that models have a large uncertainty in predicting AOD themselves. Therefore, a reliance on evaluating with only this variable is somewhat questionable. Furthermore, in the Southern Hemisphere satellites revivals are hindered persistent high levels of clouds and that dust activity also tends to occur later in the afternoon after the overpass of polar satellites (e.g., Gassó & Torres, 2019). Comparison to observations of dust concentration and deposition (e.g., Albani et al., 2015), can alleviate the dependency on remote sensing and modelling of one variable. I recommend such an evaluation be included.

The explanation for the approximate doubling of emissions in the optimal simulation appears to be to better match North African dust sources, but is the optimization not best served if undertaken regionally? For example, as D. A. Ridley is a co-author a regional AOD evaluation based on (Ridley et al., 2016) should be possible. An optimal estimation of the regional source strengths is particularly important given the goal is to prove a dataset for the community where other dust regions are important contributors to regional climate. E.g., within the Southern Hemisphere in providing IN or dusts role in marine biogeochemical cycles.

As significant amounts of dust mass occur above 6um (Adebiyi & Kok, 2020; Ryder et al., 2019), how does this impact results here and what are the impacts for generating a dataset for other CTMs given the size bins and cut off here?

Minor Comments: L103: Dust entrainment and deposition

L106: Why is a fixed global value of clay fraction used when datasets are readily available giving the regional fraction. E.g., (Journet et al., 2014)

L119: Define HEMCO?

Refs: Adebiyi, A. A., & Kok, J. F. (2020). Climate models miss most of the coarse dust in the atmosphere. Science Advances, 6(15), 1–10.

https://doi.org/10.1126/sciadv.aaz9507

Albani, S., Mahowald, N. M., Winckler, G., Anderson, R. F., Bradtmiller, L. I., Delmonte, B., et al. (2015). Tweleve thousand years of dust: the Holocene global dust cycle constrained by natural archives. Climate of the Past, 11(6), 869–2015. https://doi.org/10.5194/cp-11-869-2015

Gassó, S., & Torres, O. (2019). Temporal Characterization of Dust Activity in the Central Patagonia Desert (Years 1964–2017). Journal of Geophysical Research: Atmospheres, 124(6), 3417–3434. https://doi.org/10.1029/2018JD030209

Journet, E., Balkanski, Y., & Harrison, S. P. (2014). A new data set of soil mineralogy for dust-cycle modeling. Atmospheric Chemistry and Physics, 14(8), 3801–3816. https://doi.org/10.5194/acp-14-3801-2014

Ridley, D. A., Heald, C. L., Kok, J. F., & Zhao, C. (2016). An observationally constrained estimate of global dust aerosol optical depth. Atmospheric Chemistry and Physics, 16(23), 15097–15117. https://doi.org/10.5194/acp-16-15097-2016

Ryder, C. L., Highwood, E. J., Walser, A., Seibert, P., Philipp, A., & Weinzierl, B. (2019). Coarse and giant particles are ubiquitous in Saharan dust export regions and are radiatively significant over the Sahara. Atmospheric Chemistry and Physics, 19(24), 15353–15376. https://doi.org/10.5194/acp-19-15353-2019

---

## Author Comment (AC1) · 3 Apr 2021

**Responses to reviewers' comments**

We would like to thank the four reviewers for their comprehensive assessments and constructive comments on our manuscript. We include a point-by-point response to each comment from the submitted manuscript and *"proposed revisions in our revised manuscript"* to address the comments below. The annotated line numbers refer to the preprint version of the manuscript.

We have carefully revised the manuscript to (1) expand the evaluation of the simulation using the updated offline dust emissions against measurements of surface fine dust concentrations over the US and several independent measurements of surface dust concentration over the globe, (2) supplement more technical details of the proposed method, and (3) improve logic and clarity.

We believe our response has addressed the reviewers' comments as described in more detail below.
* * *
**Anonymous Referee #1:**
General comments:
In this paper, Meng et al. proposed a grid-independent dust emissions module for CTMs (GEOS-Chem in this case), where dust flux is being calculated a priori and offline within the emissions module (HEMCO in this case). In essence, this approach seems to be replacing the error due to interpolating meteorological fields with those due to interpolating the final dust flux. Considering nonlinear relations between meteorological fields and calculated dust fluxes, I am concerned about using this interpolated dust flux together with the interpolated meteorological fields later in a CTM to represent dust transport, diffusion, deposition, etc. The benefit of an online approach (other than the aerosol feedback) is a physical consistency between wind, relative humidity, temperature, soil moisture, etc. representing both dust emissions and other phenomena such as dust transport and deposition, and I am not convinced that the offline approach provides the same faithfulness. Along that line, I found a number of major issues both with the proposed approach as well as its evaluation as summarized in the next section.
Response: Thank you for your comment. The purpose of our manuscript may have been misunderstood. Our goal is to provide the most accurate dust emissions possible, regardless of model resolution. We have tried to more clearly motivate the manuscript. We have added text to clarify the extent of the large errors associated with existing approaches that rely on coarse resolution meteorological fields. Specifically, we have added to line 53 a clearer motivation for the offline approach *"The nonlinear dependence of dust emissions on meteorology introduces an artificial dependence of simulations upon model resolution (Ridley et al., 2013)"*, and to line 55, *"Smoothing meteorological fields to coarse resolution can lead to wind speeds falling below the emission threshold in regions that do emit dust."* and to line 62, *"Resolution-dependent mineral dust emission would vary by a factor of 3 from C360 to C24 (Ridley et al., 2013). Such large resolution-dependent biases would undermine applications of CTMs to assess dust effects, and would lead to large within-simulation inconsistency for stretched grid simulations that can span the entire resolution range simultaneously"*, and to line 195, *"The difference between the online and offline dust emissions, shown in Fig. 2f, can be considered the error in the online approach arising from coarse resolution meteorological fields."*

Specific comments:

1(a)/ If the only major benefit of the offline approach is a better resolution, simulations should be conducted with various grid resolutions rather than just contrasting the two extremes (2deg x 2.5deg vs. 0.25deg x 0.3125deg).

Response: We now emphasize that the major benefit of the offline approach is consistency across resolutions in line 283: "*The native resolution offline dust emissions facilitate consistent evaluation and application across all model resolutions*". We also note that simulations conducted in this study are all in 2x2.5 deg, which is the most commonly used GEOS-Chem simulation resolution, as stated in line 150: "We conduct global simulations with GEOS-Chem (version 12.5.0) at a horizontal resolution of 2° by 2.5° for the year 2016" in Sect. 2.4.

1(b)/ Specifically, comparisons should be conducted to ensure that the online and offline techniques indeed give the exact same dust flux with the same grid resolution.

Response: Indeed, the online and offline emissions at the same base resolution (0.25° by 0.3125°) are identical by design since the offline emissions are archived from the online code.

2/ The soil moisture (another important meteorological factor) was not mentioned when using the offline approach. Wind and soil moisture are dynamically linked and should be represented and discussed.

Response: We added to line 104 for clarification that soil moisture is indeed used for offline emissions: "*The saltation process is dependent on the critical threshold wind speed, which is determined by surface roughness, soil type and soil moisture.*"

3 (a)/ Based on the scatterplots in figures 3 and 4, the offline model seems to almost always give higher dust emissions, but figure 2(f) shows considerably lower values from the offline model over the Sahara and Sahel. Please discuss this inconsistency.

Response: The purpose of Fig. 2f was to show the spatial difference between the online and offline dust emissions with the same global dust strength and the same dust source function. Figure 3 and 4 were comparing simulations with original online dust emissions and updated offline dust emissions. The original online dust emissions had a global total dust strength of 909 Tg yr$^{-1}$; while the updated offline dust emissions were scaled to 2,000 Tg yr$^{-1}$. We consider the simulation with the original online dust emissions as the baseline simulation. The purpose of those two comparison figures is to justify the advantage of using offline dust emissions with optimized global annual dust strength in the future simulations. We have modified the caption of Figure 3 and Figure 4 as below to make it clearer.

*"The results for the simulation using the original dust emissions are shown in blue; the results for the simulation using updated dust emissions with dust strength of 2,000 Tg yr$^{-1}$ are shown in red. The best fit lines are dashed."*

3 (b)/ Also, along that line, figure 3 should include a comparison between spatial distributions of online vs offline (overlaid with the AERONET obs) AOD. Perhaps replace the DOD/AOD column and move it to the Supplement?

Response: We will add a comparison of simulated AOD overlaid with AERONET AOD between simulations with online dust emissions (Fig 2b) and offline dust emissions (Fig 2d) in the supplement as Figure S7 (Figure R1 in this response) to show the spatial difference of simulated AOD. The difference in simulated AOD reflects the differences in the emissions (Fig 2f).

[Figure]

Figure R1. Spatial distribution of simulated AOD from simulations using the online dust emissions (left column) and offline dust emissions (middle column) that were with the same updated dust source function and the same annual dust strength (909 Tg), and the AOD differences between those two simulations (right column) in different seasons. Filled circles represent the AERONET measurements included in Figure 3.

We also add description in line 213 as the following:
*"Difference maps of simulated AOD between online and offline dust emissions are shown in Figure S7."*

4/ Throughout the abstract, main text, and conclusion sections, the offline model is argued to better resolve weak dust source regions, but no evaluations are provided for these regions.
Response: We rephrased our abstract in lines 25-27 as below:
*"These updated offline dust emissions based on high resolution meteorological fields strengthen the dust emissions over relatively weak dust source regions"*

We also add evaluation with observations near weak dust source regions. We include a climatology of dust surface concentrations measurements over 1981-2000 from several independent dust measurement sites over the globe (Kok et al. 2020). We use those sites (12 in total) (Figure R2) that either in the dust belt across Northern Hemisphere or sites relatively close to the weak emission regions in the Southern Hemisphere to evaluate our dust simulation (Figure R3).

We will add Figure R3 into our main text as Figure 6. We add a new paragraph in the main text to describe the results in Figure 6:

*"Figure 6 shows the comparison of seasonal averaged modeled and measured surface dust concentrations from 12 independent sites across the globe. The simulation using the updated offline dust emissions with dust strength of 2,000 Tg yr¹ is more consistent with the observations at almost all sites. The remaining bias at sites distant from source regions, for example sites in the Southern Hemisphere and East Asia, likely reflects remaining uncertainty in representing dust deposition. Further research is needed to address remaining knowledge gaps, such as better representing the dust size distribution and deposition during transport."*

[Figure]

Figure R2. Geolocation of the 12 independent dust surface concentration sites.

[Figure]

Figure R3. Comparison of modeled and measured seasonal averaged surface dust concentrations at 12 independent globally distributed sites for the years 1981-2000. Nine sites are in the dust belt across Northern hemisphere. The remaining 3 sites are relatively close to the weak dust emission regions in Southern Hemisphere. The results for the simulation using the original dust emissions are shown in blue; the results for the simulation using updated dust emissions with dust strength of 909 Tg yr$^{-1}$ are shown in red; the results for the simulation using updated dust emissions with dust strength of 2,000 Tg yr$^{-1}$ are shown in magenta. The measurements are in black.

5/ I see no connection (and in fact no scientific merit from the physics point of view) between the scaling factor and the offline approach. The scaling is not an advantage of the offline model as described in sec. 3.3. An online model can also be scaled using the parameter "C" in Eq. (1) of the Supplement. Additionally, the paper misses the justification behind the chosen scaling factors as well as a detailed evaluation.

Response:

We modified text starting from line 58 to clarify the connection between scaling factor and the offline approach: *"Addressing this nonlinearity is especially important for the next generation of chemistry transport models that is emerging with nimble capability for a variety of resolutions at the global scale. For example, the high performance version of GEOS-Chem (GCHP) (Eastham et al., 2018) currently offers simulation resolutions that vary by over a factor of over 100 from C24 (~ 4°x4°) to C360 (~0.25°x0.25°), with progress toward even finer resolution and toward a variable stretched grid capability (Bindle et al., 2020). Resolution-dependent mineral dust emission would vary by a factor of 3 from C360 to C24 (Ridley et al., 2013). Such large resolution-dependent biases would undermine applications of CTMs to assess dust effects, and would lead to large within-simulation inconsistency for stretched grid simulations that can span the entire resolution range simultaneously. Grid-independent high resolution dust emissions offer a potential solution to this issue."*

We also add more evaluations in our main text against other types of observations, including measurements of surface fine dust concentration over the US (Figure R4) and several independent measurements of surface dust concentration over the globe (Figure R3).

[Figure]

Figure R4. Annual and seasonal mean simulated fine dust concentrations from GEOS-Chem simulations with different dust emissions for 2016, and IMPROVE fine dust measurements, which are shown as filled circles. The right column shows the corresponding scatter plot with root mean square error (E), correlation coefficient(R) and slope (M) calculated with reduced major axis linear regression. The results for the simulation using the original dust emissions are shown in blue (left column); the results for the simulation using updated dust emissions with dust strength of 909 Tg yr$^{-1}$ are shown in red (second column); the results for the simulation using updated dust emissions with dust strength of 2,000 Tg yr$^{-1}$ are shown in magenta (third column); the right column is the sensitivity simulation with North America dust emission reduced by 30%.

6/ Additional computational time required for calculating dust fluxes in HEMCO when using the offline approach should be presented and discussed.

Response: We added the computational time required for calculating the high resolution offline dust emissions in the main text in line 126 as below:

*"The computational time required for calculating offline dust emission fluxes at 0.25° x 0.3125° resolution is around 6 hours for one-year of offline dust emissions on a compute node with 32 cores on 2 Intel CPUs at 2.1 GHz."*

**Anonymous Referee #2:**

This work provides a quick and smart way to fix an issue associated with the nonlinear dependency of soil dust emissions on grid resolutions. Dust emissions are known to be a function of fourth or third power of grid-scale wind speeds, which are not a conserved quantity while regridding meteorological data for downscaling or upscaling simulations. As a result, this causes several issues, including not only simulated discrepancy from the observations but also more seriously a breach of mass conservation of soil dust aerosols using the same meteorology but different grid resolutions. Authors developed an offline approach to pre-calculate dust emissions with the finest wind speed data and to use it independent on the model grid resolutions. By doing this, they were able to use consistent dust emissions for simulations with different grid resolutions and also even scale it to better match with observations. I recommend this work for publication at GMD after addressing a few concerns about this work as follows:
Response: Thank you for your assessment.

1. Description of soil dust mobilization using the finest wind speed data is clearly written but its application for the coarse model resolutions needs a bit of more elaboration. For example, dust emissions on a finer resolution are simply summed up online while used for coarse model simulations. If this is the case, an instantaneous mixing of dust emission would occur. I guess that this would not be a serious issue for fine-mode dust bins. But, for example, coarse-mode dust aerosols emitted from 0.2 degree grid box could be instantaneously mixed into 2 degree grid box. Any problem with this?
Response: It's a common practice for coarse resolution models to represent processes at their resolution. This process fulfills the objective of this work.

2. Following up the first comment above, how do you simulate the dry deposition of dust aerosols, which are also dependent on grid-scale mixing and other micro-scale meteorological parameters. A similar issue could arise if authors use the same gridscale dry deposition calculation especially for coarse mode dust aerosols.
Response: The dry deposition of dust includes the effect of gravitational settling and turbulent resistance to the surface. The parameterization of those two effects in GEOS-Chem is represented by the dry deposition velocity induced by those two effects. Dry deposition velocities were calculated for each bin. Those processes are done in the simulation grid. Nonlinear deposition effects are beyond the scope of this work.

We have included the following details in line 146 to elaborate the dry deposition scheme in our model.
"*Dry deposition of dust includes the effects of gravitational settling and turbulent resistance to the surface, which are represented with deposition velocities in the parameterization, implemented into GEOS-Chem by Fairlie et al. (2007).*"

3. I wonder how the model evaluation against AOD observations could properly represent the model capability to simulate dust aerosols and especially dust mobilization. Not only AOD but also Angstrom exponent or other aerosol optical properties would be helpful to give a proper measure of the model performance.
Response: Given the paucity of direct dust mass measurements at the global scale, we use AOD close to dust emissions source regions to optimize the AOD simulation. Although we do not have access to appropriate aerosol size or other optical measurements to evaluate our simulation, we conducted other comparisons with measurements of surface fine dust concentration over the US (Figure R4) and dust surface concentration measurements at several independent sites over the globe (Figure R3).

**Anonymous Referee #3:**

The authors generate high spatial and temporal resolution mineral dust emissions fields that can be prescribed for use in lower-resolution simulations with the GEOS-Chem model. Online dust emissions are well known to depend on model resolution because of nonlinearity in the governing parameterization. The use of consistent dust emissions across model resolutions overcomes this problem. I disagree with Referee 1 on his/her main point, and agree with Referee 3 on his/hers. The modeler wants to represent the most accurate dust emissions distribution possible irrespective of model resolution, and in particular whether or not they are consistent with the coarse model wind field. Representing high-resolution emissions, even crudely at lower resolution, is much preferred than not representing them at all. Specifically, smoothing of the wind fields at coarser resolution leads to wind speeds falling below the threshold and zero dust emissions in locations that do emit dust. Scaling of global emissions to match those generated at higher resolution leads to unrealistic amplification (hotspots) elsewhere. This is a real problem that the proposed methodology alleviates. The authors make good use of AERONET, MODIS-DB and MAIAC datasets to justify an annual global emission total for the year concerned. The manuscript is well written and mostly clear. I recommend the manuscript be accepted with some clarifications.

Response: Thank you very much for your assessment.

1. Question:
1.1 Please clarify how the high resolution (0.25 deg. x 0.25 deg.) satellite identified dust source function (line 79, 110) is obtained.
Response: The source function S1 provides erodibility factor for sparsely vegetated surface with a potential for accumulated fine sediments. The potential location of accumulated sediments S1 has been determined by comparing the elevation of any 1 x 1 degree grid point with its surrounding hydrological basin using the same equation 1 of Ginoux et al. (2001). The function S1 is then linearly interpolated on a 0.25 degree Cartesian grid and multiplied by the fraction of bare surface within the grid cell. Such surfaces are obtained globally from  the classes 8 (bare ground) and 9 (shrubs and bare ground) of the land cover inventory retrieved from the multi-year 8 km AVHRR data (DeFries et al., 2000). It is assumed that 100% of class 8 is bare, while only 20% for class 9. According to the survey  by the Chinese Academy of Sciences (CAS, 1998), desertification has increased the bare sandy lands in China. To include these barren lands, we follow the methodology followed by Gong et al. (2003) to use the Chinese Desertification Map (Sunling Gong personal communication) and consider the desertification classes 1 (serious desertification soil), 2 (heavy desertification soil), 3 (current desertification soil) with 80% erodible bare surface, and classes 4 (potential desertification soil) and 5 (low-grade desertification soil) with 60% erodible bare surface.

We added a new section "S2. Description of the updated source function" in the supplemental material as the following:
*"S2. Description of the updated source function*
*The updated source function provides erodibility factor for sparsely vegetated surface with a potential for accumulated fine sediments. The potential location of accumulated sediments has been determined by comparing the elevation of any 1 x 1 degree grid point with its surrounding hydrological basin using the same equation 1 of Ginoux et al. (2001). The updated source function is then linearly interpolated on a 0.25 degree Cartesian grid and multiplied by the fraction of bare surface within the grid cell. Such surfaces are obtained globally from  the classes 8 (bare ground) and 9 (shrubs and bare ground) of the land cover inventory retrieved*

*from the multi-year 8 km AVHRR data (Defries et al., 2000). It is assumed that 100% of class 8 is bare, while only 20% for class 9. According to the survey by the Chinese Academy of Sciences (CAS, 1998), desertification has increased the bare sandy lands in China. To include these barren lands, we follow the methodology followed by Gong et al. (2003) to use the Chinese Desertification Map (Sunling Gong personal communication) and consider the desertification classes 1 (serious desertification soil), 2 (heavy desertification soil), 3 (current desertification soil) with 80% erodible bare surface, and classes 4 (potential desertification soil) and 5 (low-grade desertification soil) with 60% erodible bare surface."*

1.2 Section S1 refers to Ginoux et al (2001) and Zender et al. (2003). However, how are the surface factors S and in particular the Am factors obtained at this higher resolution?
Response: S is the source function. The high resolution version was obtained as described above in the answer 1.1. We did not update Am factors in this study.

1.3 How is the updated source function then applied (presumably interpolated?) for online 2x2.5 deg. simulations? Please clarify.
Response: The updated high resolution dust source function was interpolated to 2x2.5 deg in the online 2x2.5 deg. simulations.
We have modified the text in line 154 as the following to clarify it:
*"The other one is with the updated version of source function, in which the updated fine resolution source function is interpolated to 2° by 2.5° resolution."*

2. Line 127: sounds like the default emissions would be at the 2 x 2.5 deg. resolution original source function. Please clarify.
Response: The default emission is at the 2x2.5 deg. resolution with the original source function. The modification in the text in line 153 is:
*"The first one is with the original version of dust source function, thereafter noted as original online dust simulation"*

3. Line 154: it is unclear if these 2 simulations are both conducted at 2x2.5 deg. resolution.
Response: The two simulations are both conducted at 2x2.5 deg resolution.
We stated in line 150 as: "We conduct global simulations with GEOS-Chem (version 12.5.0) at a horizontal resolution of 2° by 2.5° for the year 2016."

**Anonymous Referee #4:**

The production of an improved dust emission dataset would be useful for CTMs varying spatial resolution (and idealized simulations in multi-model studies like AeroCom). Generation of high spatial and temporal resolution mineral dust emissions fields for use in lower-resolution simulations is thus a worthwhile product. However, I find the light evaluation of the simulations a concern. Improving the evaluation would increase confidence if the proposed dust emissions are representative at both global and regional levels. For example, only having a single AERONET observation site (and only in spring?) for all of Asia is particularly concerning, especially given issues with satellite retrievals over land. There also appears to be no evaluation of Southern Hemisphere sources. Until a more comprehensive evaluation of the dust atmospheric state is undertaken an evaluation of emissions, and thus the dataset as whole, is difficult to judge.

Response: Thank you for your assessment. We have included more evaluations using different types of measurements, including fine dust concentration measurements over the US from the IMPROVE network and a compilation of surface dust concentrations measurements over 12 independent sites over the globe. Please see our responses to the specific comments.

It is probably good to point out that models have a large uncertainty in predicting AOD themselves. Therefore, a reliance on evaluating with only this variable is somewhat questionable. Furthermore, in the Southern Hemisphere satellites revivals are hindered persistent high levels of clouds and that dust activity also tends to occur later in the afternoon after the overpass of polar satellites (e.g., Gassó & Torres, 2019). Comparison to observations of dust concentration and deposition (e.g., Albani et al., 2015), can alleviate the dependency on remote sensing and modelling of one variable. I recommend such an evaluation be included.

Response:
We have added model evaluations with measurements of surface dust concentrations.

Firstly, we compare our modeled fine dust surface concentration with the observation from IMPROVE network. As shown in Figure R4, the simulations using the updated offline dust emissions can better represent the observed surface fine dust concentration measurements than the simulation using the original online dust emissions with higher correlations and slopes, annually and seasonally. Scaling the annual dust strength to 2,000 Tg/yr further improved the performance of the model simulation of fine dust concentrations in all seasons except winter. This gives us the same conclusion as the AOD evaluations.

We will add Figure R4 into our main text as Figure 5 and add a new subsection (Sect. 3.3) in the main text in line 250 to describe the results in Figure 5:
*"3.3 Evaluation of the simulations against surface dust concentration measurements*
*We also evaluate our simulations using different dust emissions against measurements of surface dust concentrations. Figure 5 shows the comparison of modeled fine dust surface concentration against the fine dust concentration observation from the IMPROVE network. The simulations using the updated offline dust emissions can better represent the observed surface fine dust concentration measurements than the simulation using the original online dust emissions with higher correlations and slopes across all seasons. Annually, the correlation between the simulation and observation increases from 0.39 to 0.68 , and the slope increases from 0.31 to 0.71 when using the updated offline dust emissions with annual dust strength of 909 Tg compared to the simulation using the original online dust emissions. Scaling the annual dust strength to 2,000 Tg/yr marginally improves the performance of the model simulation of fine dust*

*concentrations in all seasons except winter, during which the surface fine dust concentrations are overestimated. Given the specificity and density of the dust measurements, and the disconnect of North American dust emissions from the global source, we conduct an additional sensitivity simulation with North American dust emissions reduced by 30%. The right column shows that the annual slope in the resultant simulation versus observations improves to 1.07. Future efforts should focus on better representing the seasonal variation of dust emissions."*

[Figure]

Figure R4. Annual and seasonal mean simulated fine dust concentrations from GEOS-Chem simulations with different dust emissions for 2016, and IMPROVE fine dust measurements, which are shown as filled circles. The right column shows the corresponding scatter plot with root mean square error (E), correlation coefficient(R) and slope (M) calculated with reduced major axis linear regression. The results for the simulation using the original dust emissions are shown in blue (left column); the results for the simulation using updated dust emissions with dust strength of 909 Tg yr$^{-1}$ are shown in red (second column); the results for the simulation using updated dust emissions with dust strength of 2,000 Tg yr$^{-1}$ are shown in magenta (third column); the right column is the sensitivity simulation with North America dust emission reduced by 30%.

Second, we compare our modeled surface dust concentration with a compilation of surface dust concentrations measurements from 12 independent sites across the globe as shown in Figure R2

(we will include this figure in our supplement text as Figure S1). Figure R3 shows the comparison of seasonal averaged modeled and measured surface dust concentrations. The simulation using the updated offline dust emissions with dust strength of 2,000 Tg yr[1] is closer to the observations at almost all sites.

[Figure]

Figure R2. Geolocation of the 12 independent dust surface concentration sites.

We will add Figure R3 into our main text as Figure 6. We add a new paragraph in the main text to describe the results in Figure 6:

*"Figure 6 shows the comparison of seasonal averaged modeled and measured surface dust concentrations from 12 independent sites across the globe. The simulation using the updated offline dust emissions with dust strength of 2,000 Tg yr[1] is more consistent with the observations at almost all sites. The remaining bias at sites distant from source regions, for example sites in the Southern Hemisphere and East Asia, likely reflects remaining uncertainty in representing dust deposition. Further research is needed to address remaining knowledge gaps, such as better representing the dust size distribution and deposition during transport."*

[Figure]

Figure R3. Comparison of modeled and measured seasonal averaged surface dust concentrations at 12 independent globally distributed sites for the years 1981-2000. Nine sites are in the dust belt across Northern hemisphere. The remaining 3 sites are relatively close to the weak dust emission regions in Southern Hemisphere. The results for the simulation using the original dust emissions are shown in blue; the results for the simulation using updated dust emissions with dust strength of 909 Tg yr[-1] are shown in red; the results for the simulation using updated dust emissions with dust strength of 2,000 Tg yr[-1] are shown in magenta. The measurements are in black.

We updated the section 2.1, description of observations, in the main text by adding the descriptions of IMPROVE fine dust observations and the compilation of independent surface dust concentrations measurements in line 99:

*"We use ground-based surface fine dust concentration measurements over the US from the Interagency Monitoring of Protected Visual Environments (IMPROVE, http://views.cira.colostate.edu/fed/DataWizard/) network. The IMPROVE network provided 24-hr average fine dust concentrations data every third day over the national parks in the United States. We also include a climatology of dust surface concentrations measurements over 1981-2000 from 27 independent dust measurement sites over the globe (Kok et al. 2020). We use those sites (12 in total) (Figure S1) that either in the dust belt across Northern Hemisphere or sites relatively close to the weak emission regions in the Southern Hemisphere to evaluate our dust simulation"*

The explanation for the approximate doubling of emissions in the optimal simulation appears to be to better match North African dust sources, but is the optimization not best served if undertaken regionally? For example, as D. A. Ridley is a co-author a regional AOD evaluation based on (Ridley et al., 2016) should be possible. An optimal estimation of the regional source strengths is particularly important given the goal is to prove a dataset for the community where other dust regions are important contributors to regional climate. E.g., within the Southern Hemisphere in providing IN or dusts role in marine biogeochemical cycles.

Response: Interesting idea. We add a regional optimization for North America where sufficient measurements exist to optimize the reginal source (the last column in Figure R4). This idea should be developed in further research. Our current work is mostly about to introduce this offline dust emission capability to ease the nonlinear dependence of dust emission parameterizations upon model resolutions when using online dust emissions.

As significant amounts of dust mass occur above 6um (Adebiyi & Kok, 2020; Ryder et al., 2019), how does this impact results here and what are the impacts for generating a dataset for other CTMs given the size bins and cut off here?

Response: This is a very good question. As Adebiyi & Kok (2020) concluded, most current models missed the coarse dust particles (diameters beyond 5 microns). So does GEOS-Chem, the model in this study. Since we are optimizing the total dust loading by using AOD measurements, the missing of coarse dust will lead to an underestimation of the total dust emission, which exists large uncertainty in current dust emission parameterizations. However, this is more of a dust size distribution issue. It will be improved with the advanced modeling development of the dust size distribution.

As for the impacts for other CTMs, it would be similar to the impacts on GEOS-Chem, because most other CTMs also missed the coarse dust with a similar dust size cutoff. The future studies would implement coarse dust and more realistic dust size distributions in all CTMs.

Minor Comments: L103: Dust entrainment and deposition
Response: Corrected.

L106: Why is a fixed global value of clay fraction used when datasets are readily available giving the regional fraction. E.g., (Journet et al., 2014)

Response: Firstly, previous studies using the same dust mobilization scheme (DEAD) as our work have indicated that DEAD is overly sensitive to clay fraction (Zender et al., 2003) and using a global map of clay fraction does not improve the dust simulation in a global scale (Ridley et al., 2012) (see figure 11 in this reference). Second, the focus of our work is to develop the offline dust emission method to alleviate the nonlinear dependence of dust emission parameterizations upon model resolutions. So, we would like to keep as many parameters the same as the original online dust simulation as possible for the offline dust emission calculation. All in all, we use a global fixed value of clay fraction in our dust mobilization scheme as suggested by Zender er al. (2003) and Ridley et al. (2012). Implementing a clay fraction map would be an interesting research topic for future studies.

L119: Define HEMCO?

Response: We have modified our text in line 119 as the following:

*"calculate hourly emissions using the Harmonized Emissions Component (HEMCO) module described below."*

Refs:

Adebiyi, A. A., & Kok, J. F. (2020). Climate models miss most of the coarse dust in the atmosphere. Science Advances, 6(15), 1–10. https://doi.org/10.1126/sciadv.aaz9507

Albani, S., Mahowald, N. M., Winckler, G., Anderson, R. F., Bradtmiller, L. I., Delmonte, B., et al. (2015). Twelve thousand years of dust: the Holocene global dust cycle constrained by natural archives. Climate of the Past, 11(6), 869–2015. https://doi.org/10.5194/cp-11-869-2015

Gassó, S., & Torres, O. (2019). Temporal Characterization of Dust Activity in the Central Patagonia Desert (Years 1964–2017). Journal of Geophysical Research: Atmospheres, 124(6), 3417–3434. https://doi.org/10.1029/2018JD030209

Journet, E., Balkanski, Y., & Harrison, S. P. (2014). A new data set of soil mineralogy for dust-cycle modeling. Atmospheric Chemistry and Physics, 14(8), 3801–3816. https://doi.org/10.5194/acp-14-3801-2014

Ridley, D. A., Heald, C. L., Kok, J. F., & Zhao, C. (2016). An observationally constrained estimate of global dust aerosol optical depth. Atmospheric Chemistry and Physics, 16(23), 15097–15117. https://doi.org/10.5194/acp-16-15097-2016

Ryder, C. L., Highwood, E. J., Walser, A., Seibert, P., Philipp, A., &Weinzierl, B. (2019). Coarse and giant particles are ubiquitous in Saharan dust export regions and are radiatively significant over the Sahara. Atmospheric Chemistry and Physics, 19(24), 15353–15376. https://doi.org/10.5194/acp-19-15353-2019

**References for our responses:**

Bindle, L., Martin, R. V., Cooper, M. J., Lundgren, E. W., Eastham, S. D., Auer, B. M., Clune, T. L., Weng, H., Lin, J., Murray, L. T., Meng, J., Keller, C. A., Pawson, S., and Jacob, D. J.: Grid-Stretching Capability for the GEOS-Chem 13.0.0 Atmospheric Chemistry Model, 1–21, https://doi.org/10.5194/gmd-2020-398, 2020.

Chinese Academy of Sciences, Chinese Desertification Map, Resource and Environment Database, Beijing, 1998.

Defries, R. S., Hansen, M. C., and Townshend, J. R. G.: Global continuous fields of vegetation characteristics: A linear mixture model applied to multi-year 8 km AVHRR data, 21, 1389–1414, https://doi.org/10.1080/014311600210236, 2000.

Eastham, S. D., Long, M. S., Keller, C. A., Lundgren, E., Yantosca, R. M., Zhuang, J., Li, C., Lee, C. J., Yannetti, M., Auer, B. M., Clune, T. L., Kouatchou, J., Putman, W. M., Thompson, M. A., Trayanov, A. L., Molod, A. M., Martin, R. V., and Jacob, D. J.: GEOS-Chem High Performance (GCHP v11-02c): a next-generation implementation of the GEOS-Chem chemical transport model for massively parallel applications, 11, 2941–2953, https://doi.org/10.5194/gmd-11-2941-2018, 2018.

Fairlie, T. D., Jacob, D. J., and Park, R. J.: The impact of transpacific transport of mineral dust in the United States, Atmospheric Environment, 41, 1251–1266, https://doi.org/10.1016/j.atmosenv.2006.09.048, 2007.

Ginoux, P., Chin, M., Tegen, I., Prospero, J. M., Holben, B., Dubovik, O., and Lin, S.-J.: Sources and distributions of dust aerosols simulated with the GOCART model, 106, 20255–20273, https://doi.org/10.1029/2000JD000053, 2001.

Gong, S. L., Zhang, X. Y., Zhao, T. L., McKendry, I. G., Jaffe, D. A., and Lu, N. M.: Characterization of soil dust aerosol in China and its transport and distribution during 2001 ACE-Asia: 2. Model simulation and validation, 108, https://doi.org/10.1029/2002JD002633, 2003.

Kok, J. F., Adebiyi, A. A., Albani, S., Balkanski, Y., Checa-Garcia, R., Chin, M., Colarco, P. R., Hamilton, D. S., Huang, Y., Ito, A., Klose, M., Leung, D. M., Li, L., Mahowald, N. M., Miller, R. L., Obiso, V., Pérez García-Pando, C., Rocha-Lima, A., Wan, J. S., and Whicker, C. A.: Improved representation of the global dust cycle using observational constraints on dust properties and abundance, 1–45, https://doi.org/10.5194/acp-2020-1131, 2020.

Ridley, D. A., Heald, C. L., and Ford, B.: North African dust export and deposition: A satellite and model perspective, Journal of Geophysical Research: Atmospheres, 117, https://doi.org/10.1029/2011JD016794, 2012.

Zender, C. S., Bian, H., and Newman, D.: Mineral Dust Entrainment and Deposition (DEAD) model: Description and 1990s dust climatology, 108, https://doi.org/10.1029/2002JD002775, 2003.

---

## Author Response (AR2)

**Responses to reviewers' comments**

Dear Editor:

We would like to thank the two reviewers for their additional comments on our manuscript. We include a point-by-point response to each comment and *"revisions in the revised manuscript"* to address the comments below. The annotated line numbers refer to the revised version of the manuscript with markups along with this response.

We believe our response has addressed the reviewers' comments as described in more detail below.

Yours sincerely,

Jun Meng

On behalf of the authors
* * *
**Anonymous Referee #4 (Report 1):**

The Authors have addressed my main concern that the observational comparison relied on a global AOD comparison by including additional analysis, including dust concentration data. The evaluation has thus been improved and the new Figures are good.
Response: Thanks for the assessment.

The secondary comment regarding the benefits of a regional constraint over a global one have has been partially addressed with the inclusion of the North American regional study and an assessment of how varying the dust emission magnitude by +/- 500Tg around the proposed 2000Tg flux impacts regional comparisons. I agree there is more work here to be undertaken in the future here. A small additional discussion of what regional refinements could be undertaken in the future and what observations are missing/needed is thus beneficial in place of such an analysis.
Response: We have add texts in section 3.4 to discuss more about the future work on dust emissions.
Lines 354-356: *"More dust-specific observations are needed to constrain dust emissions for the Asian deserts region and other deserts."*

Also, in Section 3.4 it is stated:
"Although the central Asian deserts and regions with AERONET observations (Fig. S10) are better represented by the simulation with global total annual dust emission scaled to 2,500 Tg." Is there a reason a harmonized emission dataset was not produced which, for example, would include emissions from East Asia from this increased emissions dataset, while retaining the baseline from the other at 2000Tg?

Response: A lack of observations over Asian deserts region, for example a dust concentration measurements network, is the reason why we did not provide a dust strength scale factor for this region at this stage.
We added in the text in lines 353-354 : *"We refrain from applying a regional scale factor to the central Asian deserts given the paucity of in situ measurements."*

Minor Comments:
Throughout: Suggest changing "Sahara" for "North Africa"
Response: Great suggestion. We have replaced "Sahara" with "North Africa" for texts when Middle East and Asian deserts are involved.

L47: For completeness consider adding ocean phytoplankton fertilization.
Response: We have added references for considering the ocean phytoplankton fertilization.
Line 44-45: *"on the biosphere by fertilizing the tropical forest (Bristow et al,. 2010; Tang et al., 2017) and ocean (Jickells et al., 2005; Guieu et al., 2019; Tagliabue et al., 2017)"*

L64: Factor of 16? or over an order of magnitude?
Response: The factor for varying over one dimension is around 16 (from 4° to 0.25°). The overall model horizontal resolution in two dimension will vary by over a factor of 100 (from 4° x 4° to 0.25° x 0.25°).

L187: The range of global dust emissions given at 426 – 2,726 Tg/yr for AeroCom models is not the range given in Table 3 of Huneeus et al. (2011) which is 514 – 4313 Tg/yr. Is there a reason for this discrepancy?
Response: Thanks for catching this. We were counting the sum of North Africa and Middle East dust emissions reported in the abstract of Huneeus et al. (2011) in our previous version of manuscript. We realized that the global total dust emission range in Table 3 of that reference is more appropriate to cite. We have corrected the numbers in the revised manuscript.
Line 183: *"which are in the range of the current dust emission estimates of over 514 – 4,313 Tg yr$^{-1}$ (Huneeus et al., 2011)."*

**Anonymous Referee #1 (Report 2):**

I appreciate authors' effort into revising the manuscript. I think objectives of the work are more clarified now and enhanced evaluations help with building confidence in the new model. I still have one concern that I hope the authors can address. New updates to the model comprised of two factors: 1) offline vs. online dust calculations and 2) scaling the total dust emissions to 2000 TG/yr. The manuscript's title, abstract, and conclusions seem to emphasize on the first factor (e.g., line 25: "These updated offline dust emissions based on high resolution meteorological fields strengthen dust emissions over relatively weak dust source regions, such as in southern South America, southern Africa and the southwestern United States. Identification of an appropriate dust emission strength is facilitated by the resolution independence of offline emissions."), but arguably the second factor seemed to have even more pronounced impact in a number of cases (e.g., see figure 6). I refer back the authors to my previous comment (comment # 5) that scaling can indeed be easily implemented together with an online model, so ALL the

benefits observed here should not be attributed to the offline modeling approach. I hope this makes sense and convincing to authors to conduct a fair comparison, which eventually helps with the quality of the manuscript.

Response: Thanks for the comment.

We added texts in lines 358-362: *"Although the main purpose of this manuscript is to develop and evaluate an offline grid-independent inventory, it is worth noting that online models have the capability to scale to a target source strength. In that context the global source strength identified here may be of use for global online models to scale to the global source strength, with the caveat that differences in dust parameterization, dust optics, and deposition may affect performance."*

Also as a very minor point: I suggest providing the link to the IMPROVE's main webpage when citing the network (http://vista.cira.colostate.edu/Improve).

Response: Done.

Line 104: *"We use ground-based surface fine dust concentration measurements over the US from the Interagency Monitoring of Protected Visual Environments (IMPROVE, http://vista.cira.colostate.edu/Improve/ ) network."*

**References**

Jickells, T. D., An, Z. S., Andersen, K. K., Baker, A. R., Bergametti, G., Brooks, N., Cao, J. J., Boyd, P. W., Duce, R. A., Hunter, K. A., Kawahata, H., Kubilay, N., laRoche, J., Liss, P. S., Mahowald, N., Prospero, J. M., Ridgwell, A. J., Tegen, I., and Torres, R.: Global Iron Connections Between Desert Dust, Ocean Biogeochemistry, and Climate, 308, 67–71, https://doi.org/10.1126/science.1105959, 2005.

Guieu, C., Azhar, M. A., Aumont, O., Mahowald, N., Levy, M., Ethé, C., and Lachkar, Z.: Major Impact of Dust Deposition on the Productivity of the Arabian Sea, 46, 6736–6744, https://doi.org/10.1029/2019GL082770, 2019.

Tagliabue, A., Bowie, A. R., Boyd, P. W., Buck, K. N., Johnson, K. S., and Saito, M. A.: The integral role of iron in ocean biogeochemistry, 543, 51–59, https://doi.org/10.1038/nature21058, 2017.